# Cytotoxic and Antiproliferative Effects of Diarylheptanoids Isolated from *Curcuma comosa* Rhizomes on Leukaemic Cells

**DOI:** 10.3390/molecules25225476

**Published:** 2020-11-23

**Authors:** Natsima Viriyaadhammaa, Aroonchai Saiai, Waranya Neimkhum, Wariya Nirachonkul, Wantida Chaiyana, Sawitree Chiampanichayakul, Singkome Tima, Toyonobu Usuki, Suwit Duangmano, Songyot Anuchapreeda

**Affiliations:** 1Department of Medical Technology, Faculty of Associated Medical Sciences, Chiang Mai University, Chiang Mai 50200, Thailand; fai.natsima@gmail.com (N.V.); nirachonkul.w@gmail.com (W.N.); chiampanich@gmail.com (S.C.); singkome@gmail.com (S.T.); 2Department of Chemistry, Faculty of Science, Chiang Mai University, Chiang Mai 50200, Thailand; saiai_aroonchai@hotmail.com; 3Department of Pharmaceutical Technology, Faculty of Pharmaceutical Sciences, Huachiew Chalermprakiet University, Samutprakarn 10250, Thailand; waranya.ne@gmail.com; 4Department of Pharmaceutical Science, Faculty of Pharmacy, Chiang Mai University, Chiang Mai 50200, Thailand; aa_rx105@hotmail.com; 5Cancer Research Unit of Associated Medical Sciences (AMS CRU), Faculty of Associated Medical Sciences, Chiang Mai University, Chiang Mai 50200, Thailand; 6Center for Research and Development of Natural Products for Health, Chiang Mai University, Chiang Mai 50200, Thailand; 7Department of Materials and Life Sciences, Faculty of Science and Technology, Sophia University, Tokyo 102-8554, Japan; t-usuki@sophia.ac.jp

**Keywords:** Zingiberaceae, *Curcuma comosa*, diarylheptanoids, cytotoxicity, antioxidant, anti-inflammatory, haemolysis, anticancer, Wilms’ tumour 1

## Abstract

*Curcuma comosa* belongs to the Zingiberaceae family. In this study, two natural compounds were isolated from *C. comosa*, and their structures were determined using nuclear magnetic resonance. The isolated compounds were identified as 7-(3,4-dihydroxyphenyl)-5-hydroxy-1-phenyl-(1*E*)-1-heptene (**1**) and *trans*-1,7-diphenyl-5-hydroxy-1-heptene (**2**). Compound **1** showed the strongest cytotoxicity effect against HL-60 cells, while its antioxidant and anti-inflammatory properties were stronger than those of compound **2**. Compound **1** proved to be a potent antioxidant, compared to ascorbic acid. Neither compounds had any effect on red blood cell haemolysis. Furthermore, compound **1** significantly decreased Wilms’ tumour 1 protein expression and cell proliferation in KG-1a cells. Compound **1** decreased the WT1 protein levels in a time- and dose- dependent manner. Compound **1** suppressed cell cycle at the S phase. In conclusion, compound **1** has a promising chemotherapeutic potential against leukaemia.

## 1. Introduction

*Curcuma comosa* belongs to the *Curcuma* genus. In Thailand, *C. comosa* is commonly known as “Wan Chak Motluk” and is found in the northern and south-eastern parts of the country. Its rhizomes are round, lack horizontal branches, are brown coloured on the inside and have an aromatic smell. The plant has been used as a folk medicine for women to manage unpleasant symptoms associated with the urogenital system, such as vaginal dryness, dysmenorrhoea (painful menstruation), amenorrhoea (absence of menstruation), and menorrhagia (abnormal menstruation, or too much menstruation) [1]. Moreover, the *Curcuma* genus exhibits anticancer [2], antioxidant [3], and anti-inflammatory properties [4]. It can also suppress abdominal pain associated with chronic pelvic disorders by enabling uterine contractions in the urogenital system [5]. The isolated compounds from *C. comosa* has been well identified in their structure, estrogenic activity, and osteoblast proliferation and differentiation [6,7]. The plant contains bioactive compounds with anticancer, antioxidant, or anti-inflammatory properties. Compound-092, (3*S*)-1-(3,4-dihydroxyphenyl)-7-phenyl-(6*E*)-6-hepten-3-ol from *C. comosa* demonstrates pro-oxidant activity (GSH and ROS determination) of diarylheptanoid bearing a catechol moiety in the induction of apoptosis in murine P388 leukaemia [8]. Two previously identified compounds **1** and **2** were isolated in our study [6]. Compound **1** was previously studied for its antioxidant and cytotoxicity effects on murine P388 leukaemic cells whereas compound **2** had estrogenic activity. Our study compares and demonstrates the role of F-EtOAc, F-Hex, and purified compounds of *C. comosa* in cancer cytotoxicity, red blood cell haemolysis, and antioxidant and anti-inflammatory activities. Furthermore, antiproliferation via Wilms’ tumour 1 (WT1) protein suppression following non-cytotoxic dose (at IC_20_ value) treatment was observed in the KG-1a leukaemic cell model. The WT1 protein is a leukaemic cell biological marker involved in leukaemic cell proliferation [9]. Thus, the aim of this study was to purify compounds from *C. comosa*, characterise their chemical structures and study their biological activities in leukaemic cells.

## 2. Results and Discussion

### 2.1. Fractional Extracts and Pure Compounds following Column Chromatography

In this study, 5 kg of dried rhizomes were extracted using ethyl acetate and hexane with relative polarities of 0.228 and 0.009, respectively, compared to water (1.000). The ethyl acetate fraction (F-EtOAc) and the hexane fraction (F-Hex) were obtained by maceration and evaporation. The yields of the extracts were 2.09% and 2.80%, respectively. Both fractions were separated using silica gel column chromatography, which led to the isolation of two known compounds **1** and **2**. The structures of these purified compounds were identified using 1D and 2D NMR spectroscopy and confirmed by comparison of their ^1^H and/or ^13^C NMR data with previously published data. The details of these two compounds have been demonstrated and clarified in the experimental section and Appendix A.

Compound **1** was obtained as a brown gum-like material. Its structure was suggested to be diarylheptanoid 7-(3,4-dihydroxyphenyl)-5-hydroxy-1-phenyl-(1*E*)-1-heptene (**1**), as shown in Figure 1 [6,10,11]. Compound **2** was obtained as a colourless oil. *trans*-1,7-Diphenyl-5-hydroxy-1-heptene seems to correspond to the obtained structure, as shown in Figure 2 [6,12,13]. Diarylheptanoids were isolated from the rhizomes of *C. comosa* Roxb. Some of the isolated diarylheptanoids exhibited oestrogenic activity comparable to or higher than that of the phytoestrogen genistein [6]. The phenolic diarylheptanoids 5-hydroxy-7-(4-hydroxyphenyl)-1-phenyl(1*E*)-1-heptene and 7-(3,4-dihydroxy-phenyl)-5-hydroxy-1-phenyl-(1*E*)-1-heptene were isolated from the rhizomes of *C. xanthorrhiza*. These two compounds exhibited hypolipidemic action by inhibiting hepatic triglyceride secretion [10].

### 2.2. Cytotoxicity of Fractional Extracts and Pure Compounds against Leukaemic Cells Compared to Other Cancer Cells Using MTT Assay

The cytotoxicity of F-Hex, F-EtOAc, and two purified active compounds was investigated using the MTT assay in leukaemic cell lines (K562, KG-1a, and HL-60) and compared to lung cancer (A549) and breast cancer (MCF-7) cell lines. K562, KG-1a, HL-60, A549, and MCF-7 cells were treated with F-EtOAc, compound **1** (purified active compound from F-EtOAc), PPF-Hex, or compound **2** (purified active compound from F-Hex). All extracts and purified compounds showed cytotoxic effects against all cell lines, as shown by the MTT assay. The IC_50_ values (inhibitory concentration at 50% growth) of F-EtOAc, compound **1**, F-Hex, and compound **2** on HL-60 cells were 10.79–19.93 μg/mL. The values for KG-1a and K562 cells were 18.52–53.56 and 26.67–40.52 μg/mL, respectively. The IC_50_ values of F-EtOAc, compound **1**, F-Hex, and compound **2** in A549 and MCF7 cancer cells were 31.81–63.49 and 15.26–35.70 μg/mL, respectively. However, compounds **1** and **2** reached cytotoxic values in each cell line (Figure 3). The compound **1** (4 μM) was previously reported to induce apoptosis in P388 leukaemic cells by activating caspase 3. Apoptosis cell death was characterized by chromatin condensation, formation of apoptotic bodies, DNA fragmentation, and externalization of plasma membrane phosphatidylserine [8].

### 2.3. Cytotoxicity of Fractional Extracts and Pure Compounds on PBMCs

The cytotoxicity of F-Hex, F-EtOAc, and the two purified active compounds was evaluated in peripheral blood mononuclear cells (PBMCs). PBMCs were used as a representative normal cell model for studying cytotoxicity. PBMCs were collected from volunteers (three males and two females). The IC_50_ values of F-EtOAc, compound **1**, F-Hex, and compound **2** in PBMCs were 30.91 ± 8.47, 22.40 ± 0.85, 44.38 ± 6.08, and 38.40 ± 1.50 μg/mL, respectively (Figure 4). PBMC IC_50_ values indicated that F-EtOAc, F-Hex and compound **2** from *C. comosa* were cytotoxic against PBMCs according to the National Cancer Institute reference values (25 μg/mL) [14]. However, when observing the concentration at IC_20_ values of KG-1a cells for WT1 protein expression (Western blotting), the concentrations F-EtOAc (4.58 μg/mL), compound **1** (2.30 μg/mL), F-Hex (26.96 μg/mL), and compound **2** (29.90 μg/mL) were not cytotoxic to PBMCs.

### 2.4. Antioxidant and Anti-inflammatory Activities of Compound ***1*** and Compound ***2***

To compare the antioxidant activity of compound **1** and compound **2**, the antioxidant activities of the purified compounds are shown in Table 1. Compound **1** demonstrated significantly higher antioxidant activity than compound **2** (*p* < 0.05). Interestingly, compound **1** showed potent antioxidant activities with Trolox equivalent antioxidant capacity (TEAC) and antioxidant equivalent concentration (EC_1_) values comparable to those of ascorbic acid, a widely known potent antioxidant, both directly via radical scavenging and indirectly through regeneration of other antioxidant systems [15,16]. Therefore, compound **1** was suggested as an antioxidant with potent radical-scavenging properties and ferric-reducing antioxidant powers. Compound **1** contributes to the overall antioxidant activity of *C. comosa*. Since antioxidants have the ability to reduce oxidative stress in cells, they are useful for the treatment of various conditions, such as cancer, cardiovascular diseases, gastrointestinal diseases, inflammation, and neurodegenerative diseases [17,18,19]. Compound-**092** or compound **1** in this study decreased GSH levels but did not significantly increase intracellular ROS [8].

The dose-response curve of RAW 264.7 cell viability of compounds purified from *C. comosa* extract is shown in Figure 5. The IC_20_ values, which represent the concentrations at which 80% of RAW 264.7 cells were viable, of compounds **1** and **2** were 18.47 ± 0.36 and 26.61 ± 0.40 μg/mL, respectively. Therefore, compound **2** tended to be safer for use in RAW 264.7 cells than compound **1**. The concentration at the IC_20_ value of each sample was used for further anti-inflammatory activity determination. Figure 6 illustrates the anti-inflammatory activity of the compounds purified from the *C. comosa* extract. Compound **1** showed potent inhibitory activity against both IL-6 and TNF-α. Interestingly, compound **1** exhibited a significantly more potent inhibition of IL-6 than dexamethasone (*p* < 0.05). The IC_50_ values of compound **1** against IL-6 and TNF-α, which were 3.96 ± 0.12 ng/mL and 0.94 ± 0.03 μg/mL, were almost 100 times lower than that of dexamethasone. In addition, compound **1** inhibited TNF-α inhibition unlike dexamethasone (*p* > 0.05). Meanwhile, compound **2** did not affect IL-6 and only a slight inhibitory effect on TNF-α secretion. This suggests that compound **1** is a more potent anti-inflammatory agent.

### 2.5. Effects of Fractional Extracts and Pure Compounds on Red Blood Cell Haemolysis

Haemolysis is the destruction of red blood cells, which causes the release of haemoglobin and ultimately anaemia. The effects of crude extracts, fractions, and purified compound extracts on red blood cell haemolysis should be determined prior to use. A red blood cell haemolysis assay was performed to determine the effects of F-EtOAc, compound **1**, F-Hex, and compound **2** on red blood cells at the concentration increased up to IC_50_ values of each fractional extract and pure compound. Figure 7 illustrates the effect of F-EtOAc, compound **1**, F-Hex, and compound **2** at indicated doses on red blood cell haemolysis. Interestingly, all concentrations tested with F-EtOAc, compound **1**, F-Hex, and compound **2** showed less than 5% haemolysis, suggesting that they are not haemolysis inducing agents [20].

### 2.6. Formatting Effects of Fractional Extracts and Pure Compounds on WT1 Protein Expression and Cell Proliferation

In this study, the WT1 protein was used as a biomarker of leukaemic cell proliferation and was determined using Western blotting. KG-1a cells were used as the leukaemic cell model, since this cell line has high levels of WT1 protein [21,22]. The IC_20_ values of F-EtOAc, compound **1**, F-Hex, and compound **2** (4.58, 2.30, 26.96, and 29.90 μg/mL, respectively) were used to treat and evaluate WT1 protein expression. WT1 expression following treatment decreased 64.07 ± 5.04, 60.52 ± 7.82, 52.40 ± 7.99, and 14.01 ± 11.27%, respectively, when compared to vehicle control (Figure 8A). The levels of WT1 protein following treatment with F-EtOAc, compound **1**, and F-Hex were significantly decreased and correlated with their effects on total cell numbers, as shown in Figure 8B. Compound **2** reduced total KG-1a cell number, but it did not significantly decrease WT1 protein expression. This suggests that compound **2** may target other proteins that are associated with cell proliferation. Moreover, F-Hex may contain other compounds aside from compound **2** that suppress WT1 protein expression.

### 2.7. Effects of Contact Times and Concentrations of Fractional Extracts and Pure Compounds on WT1 Protein Expression and Total Cell Numbers in KG-1a

We observed WT1 protein levels following treatment of KG-1a cells with F-EtOAc and compound **1** for 24, 48, and 72 h. F-EtOAc could significantly decrease the WT1 protein levels in a time-dependent manner by 36.55 ± 5.98% (*p* < 0.05), 94.75 ± 1.34% (*p* < 0.01), and 95.11 ± 1.30% (*p* < 0.001), as compared to vehicle control. Compound **1** could significantly decrease the WT1 protein levels in a time-dependent manner by 17.96 ± 7.62% (*p* < 0.05), 37.11 ± 4.94% (*p* < 0.01), and 56.56 ± 1.69% (*p* < 0.001), respectively as compared to the vehicle control (Figure 9A). In order to study the effects of doses of the F-EtOAc and compound **1** on WT1 protein levels, the KG-1a cells were treated with various concentrations of F-EtOAc (1.5, 3, and 4.5 μg/mL) for 48 h and compound **1** (0.5, 1.5, and 2.5 μg/mL) for 72 h. F-EtOAc significantly decreased the WT1 protein levels by 43.65 ± 5.50, 62.67 ± 4.07, and 91.96 ± 5.46%, as compared to vehicle control (*p* < 0.05) (Figure 10A). Compound **1** significantly decreased WT1 protein levels by 36.89 ± 3.12 and 50.57 ± 8.95% in response to the concentrations of 1.5 and 2.5 μg/mL, respectively, as compared to vehicle control (*p* < 0.05) (Figure 10C). Thus, WT1 protein levels significantly decreased following treated with F-EtOAc and compound **1** by a time- and dose-dependent manner when compared to the vehicle control. Furthermore, the total cell numbers were also significantly decreased by a time- and dose-dependent manner (Figure 9B and Figure 10B,D).

### 2.8. Effects of F-EtOAc and Compound ***1*** on Cell Cycle Distribution in KG-1a Cell Line Using Flow Cytometer

This experiment was to determine the effect of F-EtOAc and compound **1** on cell cycle distribution in KG-1a cells. KG-1a cells were cultured with F-EtOAc and compound **1** at the concentrations of IC_5_, IC_10_, and IC_20_ values for 48 h and assessed using flow cytometric analysis following DNA staining with PI. The flow cytometry data at 48 h are shown in Figure 11A. We observed that following treatments with F-EtOAc (1.14, 2.29, and 4.58 μg/mL, respectively) and compound **1** (0.58, 1.15, and 2.30 μg/mL, respectively), cells were significantly arrested at the S phase post F-EtOAc treatment (IC_20_ value) by 35.50 ± 3.73% (*p* < 0.05) and compound **1** treatments (IC_5_, IC_10_, and IC_20_ values) by 31.90 ± 2.98, 33.90 ± 4.37, and 35.80 ± 1.71%, as compared to vehicle control (24.6 ± 0.40%) (Figure 11B,C). Sub-G1 (peak of dead cells) was observed following F-EtOAc and compound **1** treatment for 48 h. However, the percent of cell death was less than IC_20_ values. The previous study, compound-**092** (compound **1** of this study) was investigated by the experiments of externalization of plasma membrane phosphatidylserine, caspase-3 activity, mitochondrial function, and DNA fragmentation. These experiments indicated apoptosis induction in P388 cell line [8]. Thus, we hypothesized that the biochemical pathway of apoptosis was also found in our experiment.

## 3. Materials and Methods

### 3.1. Plant Maceration

*Curcuma comosa* was harvested from Chiang Dao District, Chiang Mai Province, Thailand, in August 2018. A voucher specimen, No. 023237, was deposited at an herbarium, the Northern Research Center for Medicinal Plants, Faculty of Pharmacy, Chiang Mai University, Thailand. The herbarium specimen has been studied using traditional methods of herbarium taxonomy. Fresh rhizomes of *C. comosa* (5 kg) were peeled and dried at 50 °C. The dried rhizomes were ground to a powder and macerated in hexane for three days. The liquid portion was collected, and the residual powder was further macerated and collected three times. The liquid portions of the extraction were pooled together and filtrated. The filtrate was evaporated using a rotary evaporator (N-1000, EYELA, Shanghai, China) and subsequently dried to obtain the hexane fraction (F-Hex). The residual powder was dried in a hot air oven (45 °C) and underwent another maceration with ethyl acetate to obtain the ethyl acetate fraction (F-EtOAc).

### 3.2. Column Chromatography

Silica gel grade 60 (Merck, Darmstadt, Germany) was used as the solid phase for column chromatography. Different ratios of hexane and ethyl acetate were used as liquid phases by increasing polarity to separate different compounds. Fractions were collected in quantities of at least 8 mL in a test tube every 6–8 min. Thin layer chromatography was used to determine fractions that contain compounds. Fractions containing the main purified compounds were pooled and characterized at the Faculty of Science, Chiang Mai University, to determine chemical structures using nuclear magnetic resonance (NMR) spectroscopy (Bruker, Fällanden, Switzerland). *C. comosa* fractional extracts and main compounds were stored at −20 °C. The fractional extracts or main compounds were dissolved in DMSO to obtain the working concentration (25 mg/mL) and stored at −20 °C for later use.

Silica gel 60 was packed in a column, and F-EtOAc or PPF-Hex was added to the top of the silica gel. F-EtOAc 1.577 g was first separated. The column was eluted with Hex:EtOAc at a ratio of 1:1. Fractions (8 mL/tube) were collected and observed using thin-layer chromatography (TLC). Similar TLC patterns of each fraction were selected and pooled together. Pooled fractions were observed using TLC. The most purified pooled fractions (% yield = 32.40) were selected to do the second separation with the same procedure. The column was eluted with hexane/diethyl ether at increasing ratios of 1:1, 1:2, and 1:3. First F- hex (3.025 g) purifications were accomplished using the same procedure but different gradient elutions of hexane/ethyl acetate of 1:1, 97.5:2.5, 96.5:3.5, 95:5, 90:10, 85:15, and 80:20.

### 3.3. Structure Identification

The purity of each collected fraction was determined using TLC and ^1^H-NMR. Spectroscopic analyses characterized the structure of two pure compounds. Optical rotations were measured on an Autopol I automatic polarimeter (Rudolph Research Analytical, Hackettstown, NJ, USA) at the sodium lamp (*λ* = 589 nm) D-line and were reported as follows: [α]^T^_D_ (*c* g/100 mL, solvent). ^1^H- and ^13^C-NMR spectra were measured on an AVANCE 400 (400 MHz) spectrometer (Bruker, Fällanden, Switzerland) in deuterated chloroform (CDCl_3_; Sigma-Aldrich, St. Louis, MO, USA) and deuterated methanol (CD_3_OD; Sigma-Aldrich, St. Louis, MO, USA). ^1^H-NMR spectra were reported as follows: Chemical shift (*δ*, ppm), multiplicity (s, singlet; d, doublet; t, triplet; q, quartet; m, multiplet; br, broad), coupling constants (*J*) in Hz, integration, and assignments. ^13^C-NMR spectra were reported in terms of chemical shift (*δ*, ppm).

Compound **1** was obtained as a brown gum. The ^1^H-NMR spectrum showed two sets of aromatic protons. The signals at δ 7.33–7.14 (m, 5H) could be the five protons of a phenyl ring. Another set of aromatic protons displayed at d 6.64 (m, 2H) for two protons and d 6.50 for one proton (dd, *J* = 8.0, 2.0 Hz), which could be 1,3,4-trisubstituted aromatic rings. The two signals at d 6.37 (d, *J* = 15.8 Hz) and 6.22 (dt, *J* = 15.8, 6.8 Hz) were the trans-olefinic protons’ signals attached to the phenyl ring. The signal at δ 3.57 (m, 1H) suggested the presence of methine protons attached to hydroxy group. The ^13^C-NMR spectrum of compound **1** displayed 19 signals for twelve aromatic carbons (d 146.1, 144.2, 139.2, 135.3, 131.3, 129.4 (2C), 126.9 (2C), 120.7, 116.6, 116.3), two olefinic carbons (δ 131.4, 127.8), one methine carbon-bearing hydroxy group (d 71.2) and four methylene carbons (δ 40.6, 38.1, 32.4, 30.3), respectively. By comparison of spectroscopic data with that reported in the literature, compound **1** was suggested to be 7-(3,4-dihydroxyphenyl)-5-hydroxy-1-phenyl-(1*E*)-1-heptene [6,10,11]: ^1^H-NMR (MeOD) δ 7.33–7.14 (m, 5H), 6.64 (m, 2H), 6.50 (dd, *J* = 8.0, 2.0 Hz, 1H), 6.37 (d, *J* = 15.8 Hz, 1H), 6.22 (dt, *J* = 15.8, 6.8 Hz, 1H), 3.57 (m, 1H), 2.62 (m, 1H), 2.49 (m, 1H), 2.27 (m, 2H), 1.70 (m, 2H), 1.60 (m, 2H); ^13^C-NMR (MeOD) δ 146.1 (C), 144.2 (C), 139.2 (C), 135.3 (C), 131.4 (CH), 131.3 (CH), 129.4 (2 × CH), 127.8 (CH), 126.9 (2 × CH), 120.7 (CH), 116.6 (CH), 116.3 (CH), 71.2 (CH), 40.6 (CH_2_), 38.1 (CH_2_), 32.4 (CH_2_), 30.3 (CH_2_); ${\left[\alpha \right]}_{D}^{30}$ − 108 (c 0.37 in EtOH)

Compound **2** was obtained as a colorless oil. The ^1^H-NMR of compound **2** was similar to that of compound **1** except for the signal of aromatic protons, showing ten protons. This signal indicated that compound **2** had two phenyl rings in the structure. It was also confirmed using the ^13^C NMR, which displayed only two quaternary carbons (δ 142.2 and 137.7). By comparison of spectroscopic data with that reported in the literature, compound **2** was suggested to be *trans*-*1,7-diphenyl-5-hydroxy-1-heptene* [6,12,13]: ^1^H-NMR (CDCl_3_) δ 7.48–7.22 (m, 10H, Ar-H), 6.47 (d, *J* = 15.8 Hz, 1H), 6.29 (dt, *J* = 15.8, 6.9 Hz, 1H), 3.75 (m, 1H), 2.85 (m, 1H), 2.73 (m, 1H), 2.39 (m, 2H), 1.86 (m, 2H), 1.73 (m, 2H); ^13^C-NMR (CDCl_3_) δ 142.2 (C), 137.7 (C), 130.4 (2 × CH), 128.6 (2 × CH), 128.52 (2 × CH), 128.50 (2 × CH), 127.1 (CH), 126.1 (2 × CH), 125.9 (CH), 70.9 (CH), 39.2 (CH_2_), 37.1 (CH_2_), 32.1 (CH_2_), 29.4 (CH_2_);
${\left[\alpha \right]}_{D}^{30}$ − 1 (c 1.0 in EtOH).

### 3.4. Cell Culture

Leukaemic cell lines including K562 (human erythroid leukaemic cell line, RIKEN, Tsukuba, Ibaraki, Japan) and HL-60 (human promyelocytic leukaemic cell line, ATCC, Manassas, VA, USA) were maintained in RPMI 1640 medium containing 1mM l-glutamine, 100 units/mL penicillin, 100 μg/mL streptomycin, and supplemented with 10% fetal bovine serum (FBS). KG-1a (leukaemic stem cell-like cell line with stem cell population) was cultured in IMDM medium containing 1 mM l-glutamine, 100 units/mL penicillin, 100 μg/mL streptomycin, and supplemented with 20% FBS. A549 (human lung cancer cell line, ATCC, Manassas, VA, USA) and MCF-7 (breast cancer cell line, ATCC, Manassas, VA, USA) were cultured in DMEM medium containing 1 mM l-glutamine, 100 units/mL penicillin, 100 μg/mL streptomycin, and supplemented with 10% FBS. All cancer cell lines were incubated at 37 °C under 95% humidified and 5% CO_2_.

### 3.5. Cytotoxicity Determinations by MTT Assay

The MTT (3-(4,5-dimethylthiazol-2-yl)-2,5-diphenyltetrazolium bromide) assay was used for detecting the cytotoxicity of *C*. *comosa* extracts and purified compounds on cancer cells. The cytotoxicity of F-EtOAc, F-Hex, and main compounds was investigated using MTT assay in K562, KG-1a, HL-60, A549, and MCF-7 cell lines. K562 (1.0 × 10^5^ cells/mL), HL-60 (1.0 × 10^5^ cells/mL), KG-1a (1.5 × 10^5^ cells/mL), A549 (5.0 × 10^4^ cells/mL), and MCF-7 (5.0 × 10^4^ cells/mL) were added then incubated for 48 h. Following, a 100 μL of the medium was removed, and 15 μL of MTT dye solution was added, and cells were further incubated for 4 h. Following supernatant removal, 200 μL of DMSO was added to each well and mixed thoroughly to dissolve the purple formazan crystals. The optical density was measured using an ELISA plate reader at 578 nm with a reference wavelength at 630 nm. The percentage of surviving cells was calculated from the absorbance values of the test and control wells using the following Equation (1):(1)$$\begin{array}{cc} \%\;\mathrm{Cell}\;\mathrm{viability}\;= & \frac{\mathrm{Mean}\;\mathrm{absorbance}\;\mathrm{in}\;\mathrm{test}\;\mathrm{well}}{\mathrm{Mean}\;\mathrm{absorbance}\;\mathrm{in}\;\mathrm{vehicle}\;\mathrm{control}\;\mathrm{well}} \end{array}\times 100$$

The average percentage of cells surviving at each concentration obtained from triplicate experiments was plotted as a dose-response curve. The 50% inhibitory concentration (IC_50_) was defined as the lowest concentration that inhibited cell growth by 50% compared to the untreated control.

### 3.6. Trypan Blue Exclusion

Total cell numbers were counted using the trypan blue exclusion method. Live cells have intact membranes and can exclude trypan blue dye, whereas dead cells with compromised membranes are stained by the trypan blue dye solution. A cell suspension and 0.2% trypan blue were mixed, and viable (unstained) and dead (stained) cells were counted using a hemacytometer. The percentage of viable cells was then calculated.

### 3.7. Cytotoxicity of PBMCs

Peripheral blood mononuclear cells (PBMCs) were isolated from healthy donors using Ficoll-Hypaque density-gradient centrifugation using Lymphoprep™ solution (Axis-Shield, Oslo, Norway). The PBMCs (1 × 10^6^ cells/mL) were plated in flat-bottom 96-well plates overnight in a 5% CO_2_ incubator at 37 °C. Various concentrations (3.125, 6.25, 12.5, 25, 50, and 100 μg/mL) of *C. comosa* extracts and purified compounds were then added, and cells were incubated for 48 h. The cell survival rate was assessed utilizing the MTT colorimetric assay, as previously described.

### 3.8. Antioxidant Activities Determination

#### 3.8.1. 2,2′-Azinobis 3-ethylbenzothiazoline-6-sulphonate (ABTS) Assay

The ABTS^•+^ scavenging activity of *C. comosa* extracts and purified compounds were investigated using ABTS assay [23]. The ABTS^•+^ solution was prepared by mixing 2 mL of 7 mM ABTS solution with 3 mL of 2.45 mM potassium persulfate solution and incubated in the dark. After 24 h, the resulting ABTS^•+^ solution was diluted 1:20 in absolute ethanol. Then 20 μL of the sample was mixed with 180 μL of ABTS^•+^ solution, incubated at room temperature (25 °C) for 5 min, and measured UV absorbance at 750 nm using a microplate reader (Spectrostar Nano, BMG Labtech GmbH, Ortenberg, Germany). The results were reported as Trolox equivalent antioxidant capacity (TEAC). All experiments were conducted in triplicate.

#### 3.8.2. 2,2′-Diphenyl-1-picrylhydrazyl-hydrate (DPPH) Assay

The DPPH^•^ scavenging activity of *C. comosa* extracts and purified compounds were investigated using DPPH assay [13]. Briefly, 20 μL of the sample was mixed with 180 μL of 167 μM DPPH solution, incubated at room temperature in the dark for 30 min, and measured UV absorbance at 520 nm microplate reader (DTX880, Beckman Coulter, Fullerton, CA, USA). The scavenging activity was calculated using the following Equation (2):(2)$$\%\;{\mathrm{DPPH}}^{•}\;\mathrm{inhibition}=\frac{A-B}{A}\times 100$$
where A is a UV absorbance mixture without a sample solution, and B is a UV absorbance mixture with a sample solution. l-Ascorbic acid was used as a positive control. Dose response curve was plotted and IC_50_ value was calculated by GraphPad Prism (version 2.01, GraphPad Software, San Diego, CA, USA, https//www.graphpad.com/scientific-software/prism/). All experiments were conducted in triplicate.

#### 3.8.3. Ferric Reducing Antioxidant Power (FRAP) Assay

The ferric reducing antioxidant power of *C. comosa* extracts and purified compounds were evaluated using FRAP assay [23]. FRAP solution was freshly prepared by mixing 10 mL of 0.3 M acetate buffer pH 3.6, 1 mL of 10 mM 2,4,6 tripyridyl-*S*-triazine (TPTZ) solution in 40 mM HCl, and 1 mL of 20 mM ferric chloride. Then 20 μL of the sample was mixed with 180 μL of FRAP solution, incubated at room temperature in the dark for 5 min, and the UV absorbance was measured at 595 nm using a microplate reader (Beckman Coulter DTX880, Beckman Coulter Inc., Brea, CA, USA). The results were expressed as equivalent capacity (EC_1_). l-Ascorbic acid was used as a positive control. All experiments were conducted in triplicate.

### 3.9. Anti-inflammatory Activity Determination

Anti-inflammatory activities of *C. comosa* extracts and purified compounds were investigated employing inhibitory activities against IL-6 and TNF-α secretion [24]. The mouse monocyte macrophage RAW 264.7 cells (ATCCTIB-71, American Type Culture Collection, Manassas, VA, USA) were stimulated with lipopolysaccharide (LPS) to induce inflammatory processes. The cell incubated with LPS served as vehicle control, of which the secreted cytokines were defined as 100%, whereas non-treated RAW 264.7 cells served as a negative control. Dexamethasone, an anti-inflammatory drug, was used as a positive control.

Briefly, 1 × 10^5^ cells per well in DMEM were seeded and incubated for 24 h in a CO_2_ incubator set at 37°C and 5% CO_2_-95% air. Then 1 μL of *C. comosa* extracts or purified compounds was added. Following a 2-h incubation in a CO_2_ incubator, LPS was added to make a final concentration of 1 μg/mL. After 24 h in a CO_2_ incubator, the media were removed and centrifuged at 13,500× *g* for 10 min and ELISA analysed 100 μL of the supernatant according to the manufacturer’s protocol (R&D Systems, Minneapolis, MN, USA). The optical density was measured at 450 nm and corrected with the reference wavelength of 570 nm using a multimode detector (Beckman Coulter DTX880).

RAW 264.7 cell viability was also determined simultaneously with the ELISA using MTT assay. The supernatant was removed, and MTT was added to the cells. After 2 h in a CO_2_ incubator, the supernatant was removed, and the formazan crystals were dissolved with DMSO. The optical density was measured at 570 nm and corrected with the reference wavelength of 690 nm using a multimode detector (Beckman Coulter DTX880). IL-6 and TNF-α secretion inhibitions were calculated using the following Equation (3):(3)$$\%\;\mathrm{Cytokine}\;\mathrm{inhibition}\;=\frac{A-B}{A}\times 100$$
where A is the optical density of the mixture without the sample, while B is the optical density of the mixture with the sample. Dexamethasone was used as a positive control. All experiments were conducted in triplicate.

### 3.10. RBC Haemolysis Induction

EDTA blood samples were collected from normal subjects. Post centrifugation, RBCs were collected and washed twice with 0.9% NaCl. For RBC hemolysis induction, 1 mL of 5% RBC suspension was then incubated with F-EtOAc, Compound **1**, F-Hex, and Compound **2** at 37 °C water bath for 3 h. Triton-X 100 (0.05%) and NaCl (0.9%) were used as positive and negative controls, respectively. Following incubation, the supernatant was collected using centrifugation at 970× *g* for 5 min at room temperature (25 °C), and haemoglobin concentration was measured by spectrophoto-metry at 540 nm.

### 3.11. Western Blot Analysis

KG-1a cells were adjusted to a concentration of 1.0 × 10^5^ cells/mL and cultured with F-EtOAc, Compound **1**, F-Hex, and Compound **2** at a 20% growth inhibition (IC_20_ value) of each extract. The IC_20_ value was used according to determine gene expression level following treatment without cross effect from cell death. Following a 48-h incubation, cells were washed, and the whole protein was extracted using RIPA buffer (50 mM Tris-HCl, 150 mM NaCl, 1% Triton X-100, 0.5 mM EDTA, 0.1% SDS, and a protease inhibitor cocktail). The protein concentration was determined using the Folin-Lowry method. Twenty micrograms of each sample were loaded to 7.5% SDS-PAGE and then transferred to PVDF membranes. The membranes were shaken in PBS, pH 7.4 for 5 min. Membranes were subsequently blocked in 5% skim milk in PBS, pH 7.4 for 2 h at room temperature (25 °C). Each membrane was incubated with rabbit polyclonal anti-WT1 IgG (Santa Cruz Biotechnology, Santa Cruz, CA, USA) and rabbit polyclonal anti-human GAPDH IgG (Santa Cruz Biotechnology) at dilution of 1:1000 while shaking for 2 h. Membranes were then washed then incubated with HRP-conjugated goat anti-rabbit IgG (Invitrogen™, Rockford, IL, USA) at 1:20,000 dilution while shaking for 2 h. To detect protein bands, Luminata™ Forte Western HRP substrate (Merck, Darmstadt, Germany) was added to membranes, which were then placed onto a film cassette and exposed to X-ray film (FINE Med, Hebei, China). Densitometry was quantitated using the Quantity One 1-D Analysis software (Bio-Rad, Hercules, CA, USA). The density values of WT1 bands were normalized to GAPDH bands.

### 3.12. Statistical Analysis

Data are expressed as the mean ± standard deviation (SD) or the mean ± standard error of the mean (SEM) from triplicate samples of three independent experiments. The statistical differences between the means were determined using one-way ANOVA and student T-test. The differences were considered significant when the probability value obtained was found to be less than 0.05 (*p* < 0.05).

## 4. Conclusions

This study has identified two diarylheptanoids; both were active components in the ethyl acetate and hexane fractional extracts of *C*. *comosa*. Bioassays of diarylheptanoids against cancer cells confirmed their anti-leukaemic, antioxidant, and anti-inflammatory activities. Compound **1** is a potent antioxidant and anti-inflammatory agent against both IL-6- and TNF-α-mediated inflammation. Additionally, compound **1** showed significant suppression of WT1 protein expression and leukaemic cell proliferation. WT1 protein was decreased in a time- and dose-dependent manner. Moreover, compound **1** could arrest cell cycle distribution at the S phase. These results suggest that compound **1** has a chemotherapeutic potential against human leukaemia, particularly acute myeloblastic leukaemia (AML).

## Figures and Tables

**Figure 1 molecules-25-05476-f001:**
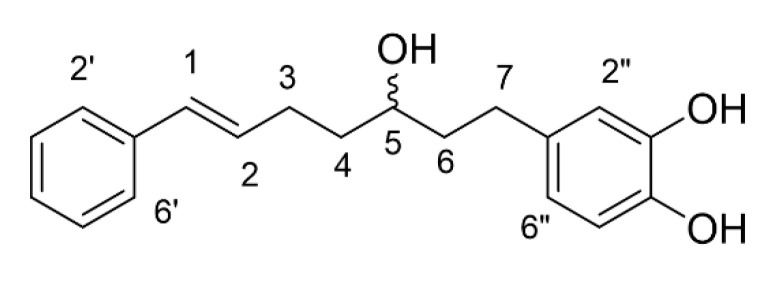
The chemical structure of compound **1** (7-(3,4-dihydroxyphenyl)-5-hydroxy-1-phenyl-(1E)-1-heptene).

**Figure 2 molecules-25-05476-f002:**
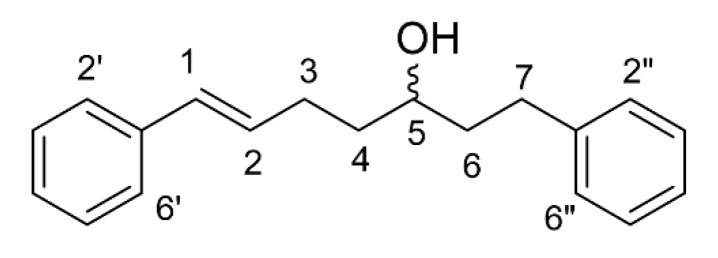
The chemical structure of compound **2** (*trans*-1,7-diphenyl-5-hydroxy-1-heptene).

**Figure 3 molecules-25-05476-f003:**
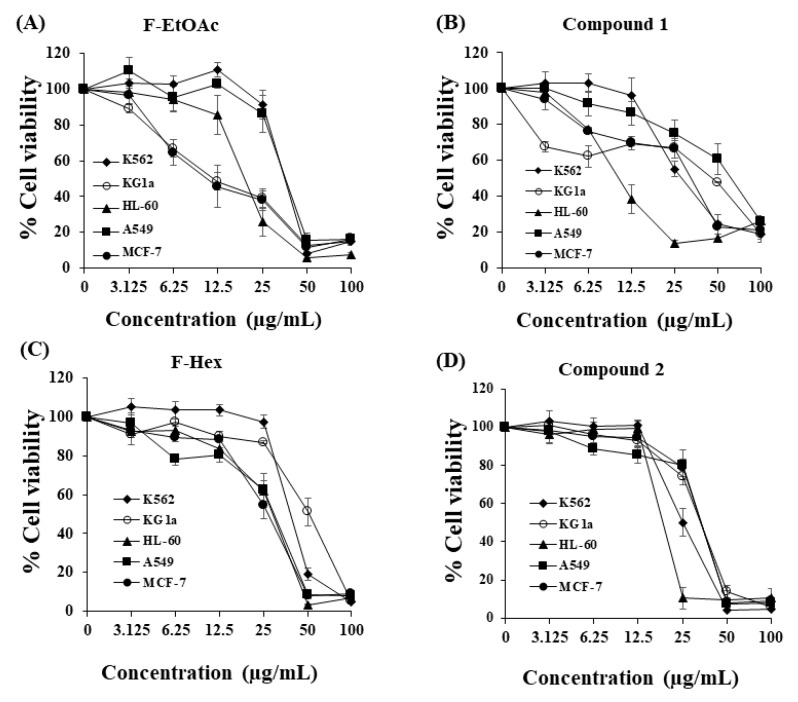
Cytotoxicity of crude fractional extracts and pure compounds from *C. comosa* to cancer cell lines. K562, KG-1a, HL-60, A549, and MCF-7 cells were treated with (**A**) ethyl acetate fraction (F-EtOAc), (**B**) compound **1**, (**C**) hexane fraction (F-Hex), or (**D**) compound **2** for 48 h. Cell viability was determined using the MTT assay. Each point represents the mean ± standard deviation (SD) of three independent experiments, each performed in triplicate.

**Figure 4 molecules-25-05476-f004:**
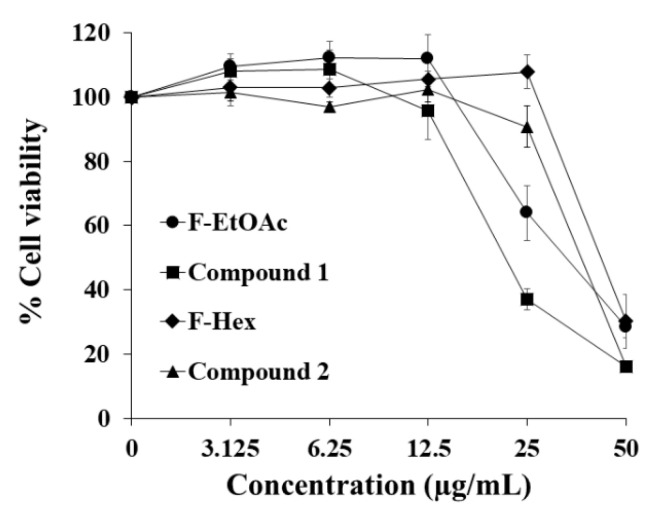
Cytotoxicity of crude fractional extracts and pure compounds from *C. comosa* to peripheral blood mononuclear cells (PBMCs). PBMCs were treated with ethyl acetate fraction (F-EtOAc), compound **1**, hexane fraction (F-Hex), and compound **2** for 48 h. The MTT assay determined the cell viability. Each point represents the mean ± SE of five independent experiments performed in triplicate.

**Figure 5 molecules-25-05476-f005:**
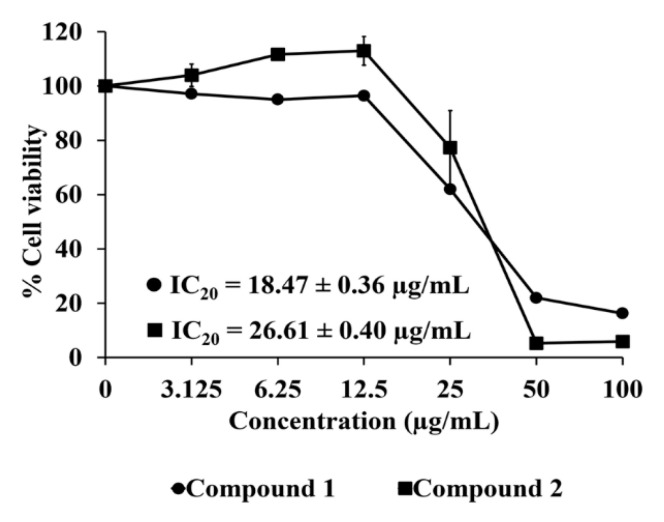
The effect of compound **1** and compound **2** on the viability of RAW 264.7 cell line using MTT assay. Each point represents the mean ± SD of three independent experiments performed in triplicate.

**Figure 6 molecules-25-05476-f006:**
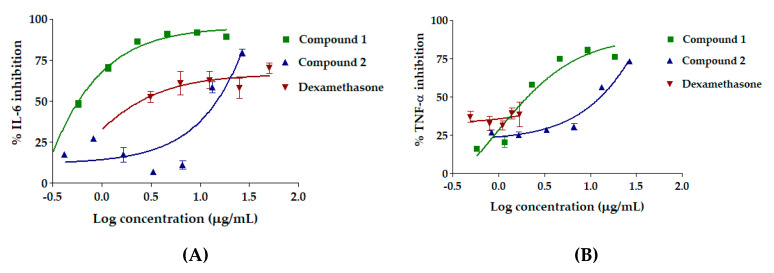
Dose response curve of dexamethasone, compound **1**, and compound **2** on (**A**) IL-6 and (**B**) TNF-α. Each curve represents the mean ± SD of three independent experiments performed in triplicate.

**Figure 7 molecules-25-05476-f007:**
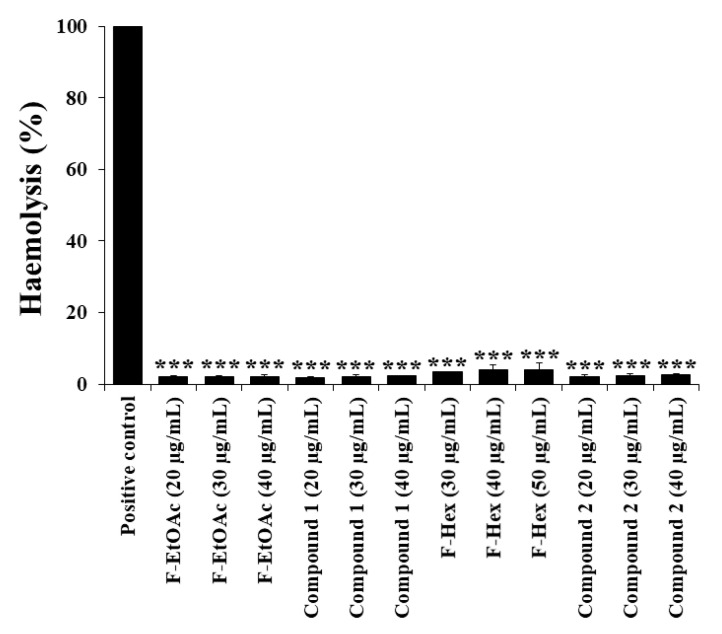
RBC hemolysis following incubation with F-EtOAc, compound **1**, F-Hex, and compound **2**. Each bar represents the mean ± SD of three independent experiments performed in triplicate. Asterisks (*) denote significant differences between C. comosa extracts and positive control (*** *p* < 0.001).

**Figure 8 molecules-25-05476-f008:**
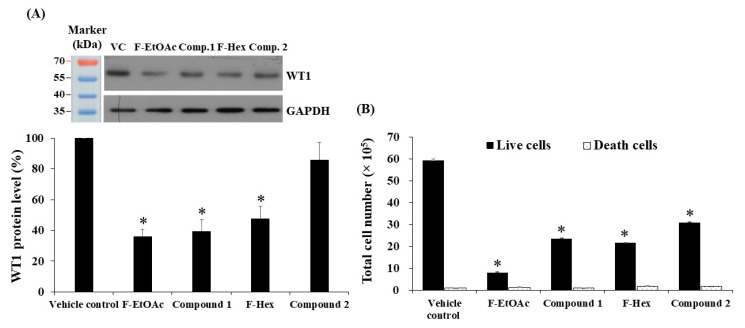
Effect of F-EtOAc, compound **1**, F-Hex, and compound **2** on KG-1a cells. (**A**) The level of WT1 protein following treatment F-EtOAc (4.58 μg/mL), compound **1** (comp. **1**; 2.30 μg/mL), F-Hex (26.29 μg/mL), or compound **2** (comp. **2**; 29.90 μg/mL) for 48 h. Protein levels were evaluated using Western blotting and analysed using scan densitometer. The levels of WT1 were normalised using glyceraldehyde phosphate dehydrogenase (GAPDH) protein levels. (**B**) Total cell number after treatment with F-EtOAc, compound **1**, F-Hex, and compound **2** for 48 h was determined via the trypan blue exclusion method. Each bar represented mean ± SD of three independent experiments performed in triplicate. Asterisks (*) denote significant differences between *C. comosa* extracts and vehicle control (* *p* < 0.05).

**Figure 9 molecules-25-05476-f009:**
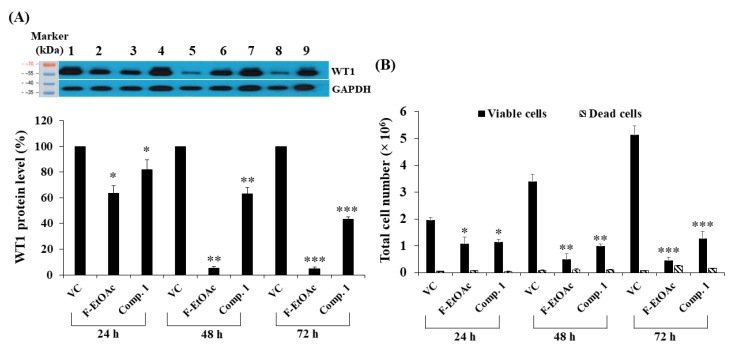
Effect of F-EtOAc and compound **1** on WT1 protein expression in KG-1a cells. (**A**) The levels of WT1 protein following treatments with F-EtOAc (4.58 μg/mL) and compound **1** (comp.1; 2.30 μg/mL) for 24, 48, and 72 h. Protein levels were evaluated using Western blotting and analysed using a scanning densitometer. The levels of WT1 were normalised using GAPDH protein levels. WT1 and GAPDH proteins following vehicle control at 24 h (No. 1), F-EtOAc at 24 h (No. 2), compound **1** at 24 h (No. 3), vehicle control at 48 h (No. 4), F-EtOAc at 48 h (No. 5), compound **1** at 48 h (No. 6), vehicle control at 72 h (No. 7), F-EtOAc at 72 h (No. 8), and compound **1** at 72 (No. 9). (**B**) Total cell number following treatment with F-EtOAc and compound **1** for 24, 48, and 72 h. Total cell numbers were determined using the trypan blue exclusion method. Each bar represented mean ± SD of three independent experiments performed in triplicate. Asterisk (*) denotes significant differences from the vehicle control; * *p* < 0.05; ** *p* < 0.01; *** *p* < 0.001.

**Figure 10 molecules-25-05476-f010:**
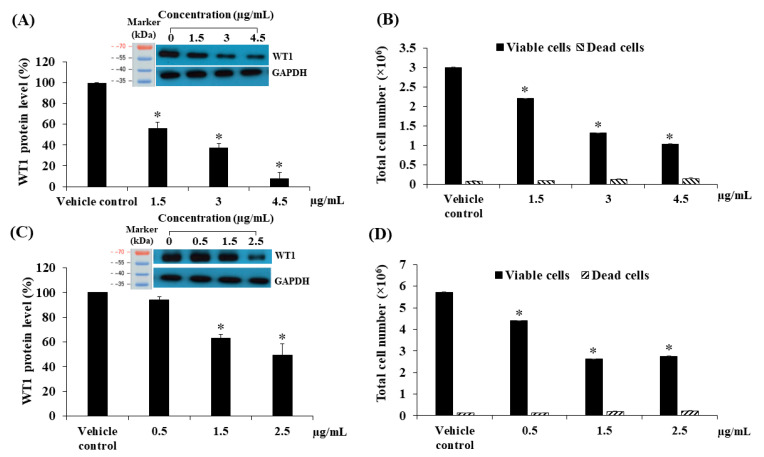
Effect of F-EtOAc and compound **1** at various concentrations on KG-1a cells. (**A**) The level of WT1 protein following treatment with F-EtOAc for 48 h. Protein levels were evaluated using western blot and analysed using a scanning densitometer. The levels of WT1 were normalised using GAPDH protein levels. (**B**) Total cell number following treatment with F-EtOAc for 48 h. Total cell numbers were determined via the trypan blue exclusion method. (**C**) The level of WT1 protein after treatment with compound **1** for 72 h. Protein levels were evaluated using Western blotting and analysed using a scanning densitometer. The levels of WT1 were normalised using GAPDH protein levels. (**D**) Total cell number following treatment with compound **1** for 72 h. Total cell numbers were determined via the trypan blue exclusion method. Each bar represented mean ± SD of three independent experiments performed in triplicate. Asterisks (*) denote significant differences from the vehicle control (* *p* < 0.01).

**Figure 11 molecules-25-05476-f011:**
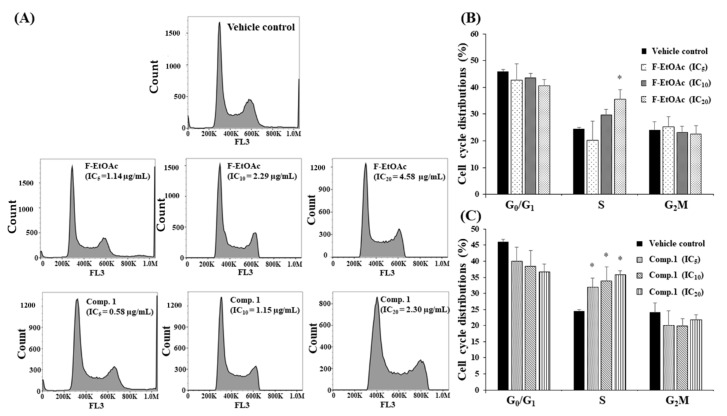
Cell cycle arrested following F-EtOAc and compound **1** treatments for 48 h. (**A**) KG-1a cells at the density of 1.5 × 10^5^ cells/mL treated with F-EtOAc or compound **1** at various inhibition concentrations (IC) from IC_5_ to IC_20_, and DMSO as vehicle control for 48 h. Cell cycle was analysed using flow cytometry. (**B**) Cell cycle distribution post F-EtOAc treatments. (**C**) Cell cycle distribution post compound **1** treatments. Data are expressed as mean ± SD of three independent experiments. Asterisks (*) denote significant differences from the vehicle control (* *p* < 0.05).

**Table 1 molecules-25-05476-t001:** Antioxidant activity of compound **1** and **2**.

Samples	TEAC (μM Torox/g Extract)	EC_1_ (mM FeSo_4_/g Extract)	IC_50_ DPPH (μg/mL)
Ascorbic acid	1.7 ± 0.2 ^a^	23.7 ± 0.7 ^a^	13.9 ± 0.5 ^a^
Compound **1**	0.9 ± 0.2 ^a^	23.2 ± 1.0 ^a^	13.0 ± 0.3 ^b^
Compound **2**	0.1 ± 0.0 ^b^	1.5 ± 0.0 ^b^	>100 ^c^

Results expressed as mean ± SD of triplicate samples. Superscript letters (a, b, and c) within the same column denote significant differences in means between different samples determined using one-way analysis of variance (ANOVA) followed by Tukey’s test (*p* < 0.05).

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
