# Peer review of "Cytotoxic and Antiproliferative Effects of Diarylheptanoids Isolated from Curcuma comosa Rhizomes on Leukaemic Cells"

_molecules, 2020, doi:10.3390/molecules25225476_

Round 1

Reviewer 1 Report

The manuscript by Viriyaadhammaa et Al. describes the cytotoxic and antiproliferative effects of the extracts and two specific components isolated from Curcuma comosa rhizomes. The article follows the line of many others who study compounds of natural origin that may have toxic activity on cancer cells. therefore, overall, the manuscript does not shine for originality.

The results show the cytotoxic effects but the biochemical pathway behind these effects is not investigated. In addition, since there is not Discussion section, no hypothesis is made about any molecules or pathways that could be involved in these effects.

Many experiments are performed using non-homogeneous concentrations and incubation times whose ratio is not adequately explained. In some points there is no precise correspondence between the results reported and the figures indicated, it seems that the organization of the manuscript had been revisited with little attention.

Specific comments:

Paragraph 2.3: Why did the authors not use non-tumor cell lines (renal, breast, etc.) instead of blood mononuclear cells as control cells?

Line 131-132: it is not clear the reference to the IC20 value and the expression of the WT1 protein without any introduction to the statement reported..

Line 151 -152: explain the acronisms TEAC and EC

Line 158: What is Compound-092?

Figure 8: Explain better why the IC20 dose was used in the second part of the experiments (Fig. 8) and insert the reference to the dose used in the legend of the figures 8 and 9. In fig. 10 instead the doses different from the previous ones are used: To explain the ratio, please.

In editing phase, the figures could be insert near the paragraph that discusses them.

Author Response

Reviewer #1

The manuscript by Viriyaadhammaa et Al. describes the cytotoxic and antiproliferative effects of the extracts and two specific components isolated from Curcuma comosa rhizomes. The article follows the line of many others who study compounds of natural origin that may have toxic activity on cancer cells. therefore, overall, the manuscript does not shine for originality.

  1. The results show the cytotoxic effects but the biochemical pathway behind these effects is not investigated. In addition, since there is not Discussion section, no hypothesis is made about any molecules or pathways that could be involved in these effects.

Answer: The biochemical pathway of diarylheptanoids from Curcuma comasa was reported by Jariyawat S. et. al. in 2011 already. Diarylheptanoids (compound-092 = compound 1 of our report), isolated from the rhizome of C. comasa, inhibited P388 (murine leukaemic cell line) cell viability by causing DNA breakage and induced apoptosis. Apoptosis cell death was characterized by chromatin condensation, the formation of apoptotic bodies, DNA fragmentation, and externalization of plasma membrane phosphatidylserine. Moreover, diarylheptanoid increased caspase-3 activity, decreased the intracellular reduced glutathione level, and impaired mitochondrial transmembrane potential. We discussed these reports on page 3, lines 109-112. However, our study tried to compare and investigate the effects of diarylheptanoids (compound 1 and compound 2) on cytotoxicity, red blood cell hemolysis, antioxidant, anti-inflammatory activities, and anticancer. Wilms’ tumour 1 (WT1) protein was used as an experimental model for this study, especially antileukaemic cell proliferation. This result exhibited that compound 1 significantly decreased Wilms’ tumour 1 protein expression and cell proliferation. It reduced the WT1 protein levels in a time- and dose- dependent manner and suppressed cell cycle at the S phase. After Wilms’ tumour 1 suppression, cells were induced to cell death.

Reference

Jariyawat, S.; Thammapratip, T.; Suksen, K.; Wanitchakool, P.; Nateewattana, J.; Chairoungdua, A.; Suksamrarn, A.; Piyachaturawat, P. Induction of apoptosis in murine leukemia by diarylheptanoids from Curcuma comosa Roxb. Cell Biol Toxicol 2011, 27, 413-423.

  1. Many experiments are performed using non-homogeneous concentrations and incubation times whose ratio is not adequately explained. In some points there is no precise correspondence between the results reported and the figures indicated, it seems that the organization of the manuscript had been revisited with little attention.

Answer: The doses and times were used differently because cell types, fractional extracts, and pure compounds showed different cytotoxicity. In this study, the IC20 values of fractional extracts and pure compounds were chosen to examine the biological activities (WT1 gene expression, antioxidant, anti-inflammatory activities, and cell cycle progression). Cytotoxicity at 20% of cell growth (IC20 value) is normally used to experiment with gene expression and cell proliferation research. The IC20 values of F-EtOAc, compound 1, F-Hex, and compound 2 were 4.58, 2.30, 26.96, and 29.90 µg/mL, respectively. The IC20 values were directly used to study the effects of fractional extracts and pure compounds on WT1 protein expressions. When study the effects of contact concentrations on WT1 protein expression, concentration was varied within IC20 values (F-EtOAc (IC20 value = 4.58 mg/ml); 1.5, 3, and 4.5 mg/ml and compound 1 (IC20 value = 2.30 mg/ml); 0.5, 1.5, and 2.5 mg/ml). However, concentration used in cell cycle progression was also within IC20 values (F-EtOAc; 1.14, 2.29, and 4.5 mg/ml and compound 1; 0.5, 1.15, and 2.5 mg/ml). The ratios were referenced from IC5, IC10, and IC20 values, respectively. However, the ratios were not different when compared to the ratios used in Western blotting. Following treatments, dose responses could be observed. In contrast, IC50 value was used in red blood cell haemolysis. Doses were varied within IC50 values. We would like to observe the cytotoxicity on normal red blood cells when the concentration increased to the IC50 value of cytotoxicity in leukemic cells.

Specific comments:

  1. Paragraph 2.3: Why did the authors not use non-tumor cell lines (renal, breast, etc.) instead of blood mononuclear cells as control cells?

Answer: In this study, we focused on the effects of fractional extracts and pure compounds on leukemic cells. Thus, the blood mononuclear cell is the best standard cell model to compare cytotoxicity with leukaemic cells. However, “PBMCs were used as a representative normal cell model for studying cytotoxicity” added on page 4, lines 136-137.

  1. Line 131-132 (lines 143-144 of revised version): it is not clear the reference to the IC20 value and the expression of the WT1 protein without any introduction to the statement reported.

Answer: We compared the concentrations at the IC20 values of F-EtOAc, compound 1, F-Hex, and compound 2 after KG-1a cell treatments (4.58, 2.30, 26.96, and 29.90 µg/mL, respectively) and PBMC treatments (22.28, 19.24, 31.49, and 27.42 µg/mL, respectively). The almost IC20 values of fractional extracts and pure compounds in KG1a cells were lower than IC20 values of PBMCs. Thus, the concentrations used for WT1 protein expression were not cytotoxic to normal blood cells. Thus, all concentrations were safe for normal cells when applied in leukaemia patients in the future. The statement and reports have been added on page 2, lines 61-62, page 4, lines 143-144, and page 13, lines 550-551.

  1. Line 151-152 (lines162-163 of revised version): explain the acronisms TEAC and EC

Answer: TEAC refers to Trolox equivalent antioxidant capacity, whereas, EC1 refers to antioxidant equivalent concentration. Both TEAC and EC1 presented the antioxidant activities of sample. More detail about TEAC and EC1 has already been provided in page number 4 lines 162-164 as following; “Interestingly, compound 1 showed potent antioxidant activities with the Trolox equivalent antioxidant capacity (TEAC) and antioxidant equivalent concentration (EC1) values comparable to ascorbic acid”.

  1. Line 158 (line 171 of revised version): What is Compound-092?

Answer: Compound-092 is the diarylheptanoids ((3S)-1-(3,4-dihydroxyphenyl)-7-phenyl-(6E)-6-hepten-3-ol) from Curcuma comasa which was reported by Jariyawat S. et. al. in 2011. Its structure is the same as compound 1. All details have been shown on page 2, lines 56-59.

  1. Figure 8: Explain better why the IC20 dose was used in the second part of the experiments (Fig. 8) and insert the reference to the dose used in the legend of the figures 8 and 9. In fig. 10 instead the doses different from the previous ones are used: To explain the ratio, please.

Answer: We explained the IC20 dose already in question No. 2 and 4. The doses were inserted already in the legends of Figures 8 and 9. Doses in Figures 8 and 9 were IC20 values. The IC20 values of F-EtOAc, compound 1, F-Hex, and compound 2 were 4.58, 2.30, 26.96, and 29.90 µg/mL, respectively. However, doses used in the figure 10 were doses within IC20 values (F-EtOAc (IC20 value = 5.11 mg/ml); 1.5, 3, and 4.5 mg/ml and compound 1 (IC20 value = 2.5 mg/ml); 0.5, 1.5, and 2.5 mg/ml). The ratios were varied with an appropriate ratio.

  1. In editing phase, the figures could be insert near the paragraph that discusses them.

Answer: The figures have been moved near the paragraph that discusses already.

Reviewer 2 Report

It is a relatively comprehensive study investigating two isolated compounds from C. comosa, and their associated bioactivities in anti-oxidant, anti-inflammatory and anti-cancer have been investigated. Several comments are shown below:

The introduction needs to be further elaborated with clearly identified knowledge gap, such as if the bioactivity of the C. comosa has been well identified and to what degree of the evidence. If the two compounds that you have isolated are novel and what has been found for their bioactivities (or none).

In line 54-55, “this study demonstrates the role of C. comosa in cancer 55 cytotoxicity, red blood cell haemolysis, and antioxidant and anti-inflammatory activities” which is misleading, as only fractions and compounds were tested? Or you actually refer to literature?

Line 155, it may not be accurate to say “C. comosa was thus a natural source of potent antioxidants” just because compound 1 showed anti-oxidant activity since overall the anti-oxidant activity of C. comosa was not determined in your study. You may say compound 1 contributes to the overall anti-oxidant activity of C. comosa.

Figure 6, would be good to have a full-dose response curve in inhibiting IL-6 and TNF within the non-cytotoxic range for compounds 1 and 2.

How is the bioactivity of the whole extract compared to compounds 1 and 2? That would determine if the isolation of compounds is necessary.

Author Response

Reviewer #2

It is a relatively comprehensive study investigating two isolated compounds from C. comosa, and their associated bioactivities in anti-oxidant, anti-inflammatory and anti-cancer have been investigated. Several comments are shown below:

  1. The introduction needs to be further elaborated with clearly identified knowledge gap, such as if the bioactivity of the C. comosa has been well identified and to what degree of the evidence. If the two compounds that you have isolated are novel and what has been found for their bioactivities (or none).

Answer: Details of isolated compounds from C. comosa were added to the introduction already as shown on page 2, lines 51-63 with red color.

  1. In line 54-55 (lines 58-60 of revised version), “this study demonstrates the role of C. comosa in cancer cytotoxicity, red blood cell haemolysis, and antioxidant and anti-inflammatory activities” which is misleading, as only fractions and compounds were tested? Or you actually refer to literature?

Answer: This sentence refers to our study. Both fractional extracts and pure compounds were tested for their biological activities. We changed “this study demonstrates the role of C. comosa …… and anti-inflammatory activities” to be “Our study demonstrates the role of C. comosa …… and anti-inflammatory activities” on page 2, lines 58-60.

  1. Line 155 (line 167 of revised version), it may not be accurate to say “C. comosa was thus a natural source of potent antioxidants” just because compound 1 showed antioxidant activity since overall the antioxidant activity of C. comosa was not determined in your study. You may say compound 1 contributes to the overall antioxidant activity of C. comosa.

Answer: The sentence “C. comosa was thus a natural source of potent antioxidants” have been changed to be “compound 1 contributes to the overall antioxidant activity of C. comosa” already on page 4, line 167-168.

  1. Figure 6, would be good to have a full-dose response curve in inhibiting IL-6 and TNF within the non-cytotoxic range for compounds 1 and 2.

Answer: We have revised the manuscript according to reviewer’s comments. The full-dose response curve in inhibiting IL-6 and TNF within the non-cytotoxic range for compounds 1 and 2 have already been provided in Figure 6 in page 5. Additionally, the IC50 values were calculated and described in page 4, lines 181-182 as following “The IC50 values of compound 1 against IL-6 and TNF-a, which were 3.96 ± 0.12 ng/mL and 0.94 ± 0.03 mg/mL, were almost 100 times lower than that of dexamethasone.”

  1. How is the bioactivity of the whole extract compared to compounds 1 and 2? That would determine if the isolation of compounds is necessary.

Answer: Whole extract (ethanolic extract) was also extracted and tested cytotoxic activity in KG-1a. However, the IC50 value of ethanolic extract is 51.35 µg/mL. The IC50 value of ethanolic extract is higher than compound 1 and compound 2 which are 43.83 and 39.66 µg/mL, respectively. Thus, compounds’ isolation is necessary to obtain more effective bioactivity than the whole extract on cell lines.

Round 2

Reviewer 1 Report

Accepted for publication